# Mindfulness Meditation and Fantasy Relaxation in a Group Setting Leads to a Diminished Sense of Self and an Increased Present Orientation

**DOI:** 10.3390/bs9080087

**Published:** 2019-08-15

**Authors:** Niko Kohls, Tobias Esch, Lea Gerber, Lucas Adrian, Marc Wittmann

**Affiliations:** 1Division of Integrative Health Promotion, Department of Social Work and Health, University of Applied Sciences Coburg, D-96450 Coburg, Germany; 2School of Medicine, Faculty of Health, Witten/Herdecke University, D-58448 Witten, Germany; 3Department of Psychology, University of Freiburg, D-79085 Freiburg, Germany; 4Institute for Frontier Areas of Psychology and Mental Health, D-79098 Freiburg, Germany

**Keywords:** mindfulness meditation, fantasy relaxation, time perception, self-awareness

## Abstract

(1) Background: Mind-body interventions (MBI), such as meditation or other relaxation techniques, have become the focus of attention in the clinical and health sciences. Differences in the effects of induction techniques are being increasingly investigated. (2) Methods: Here, we compared changes in the individual experience of time, space, and self in 44 students in an integrative health-promotion program. They participated in a study employing mindfulness meditation and a relaxation intervention with one week between sessions, thus employing a within-subjects design. (3) Results: No significant differences were detected when subjective reports were compared directly after each intervention. However, we found significant sequence effects between t1 and t2, independent of the meditation type. The sense of self diminished, the present orientation increased, and the past and future orientations decreased in both interventions. (4) Conclusions: We propose using scales to assess subjective time, self, and space as basic constituents of experience to measure the specificity of intervention methods, as well as longitudinal changes.

## 1. Introduction

In the last decades, mindfulness-meditation techniques (MMT), as well as mind-body interventions (MBI), have become a focus of attention in the clinical and health sciences [1]. MMTs and MBIs focus on the relationships among the brain, mind, body, and behavior, and their effect on health and disease [2]. For example, a 2018 systematic review of 24 Randomized Controlled Trials (RCT) studies demonstrated that MBIs have small-to-moderate positive effects on heart failure patients’ objective and subjective outcomes [3]. MBIs were also found to be able to alter the expression of our genes, alter chromosomal telomere lengths, or mitochondrial metabolism, and also potentially reduce risk for certain disease [4,5]. A 2017 review showed that MBIs can reverse certain molecular reactions, essentially generating the “opposite of the effects of chronic stress on gene expression,” which could lead to a reduced risk of inflammation-related diseases [6]. Results from a 2016 study that compared popular MBIs led researchers to conclude “mind-body interventions can improve a person’s level of mental health when compared to those who do not practice these techniques” [7]. 

A substantial amount of research has also investigated the long-term effects of MMTs and MBIs on basic psychological functions and processes, such as executive functions [8], attention regulation, cognitive flexibility [9], bistable imagery [10], and time perception [11,12,13,14,15,16]. An interesting research branch has begun focusing on the short-term effects of meditation MMTs and MBISs and underlying trait characteristics. For example, research studies have shown that a four-to-five-day meditation training unit can enhance the ability to sustain attention in a way that was previously only observed in long-term meditators [17,18]. The aim of the present study was to investigate the transitory effects on experienced states of consciousness and what changes functionally in psychological variables associated with self, space, and time perception during meditative states [19]. 

An interesting question relates to whether, when, and how different MBIs exert a specific effect on psychological functions and whether the observed effects are generic and independent of the technique employed, e.g., related to general relaxation. From the taxonomy of MBIs, it can be concluded that some techniques may influence different psychological mechanisms associated with different degrees of introspective and interoceptive processes pertaining to body schema, proprioception, and to imagination and visualization [20]. A well-known technique employing body schemata and proprioception is mindfulness breathing meditation focusing on the embodied present moment experience, while guided imagery is more associated with visualization and mentalization.

The techniques associated with mindfulness meditation are probably the most frequently studied. As part of this method, the “body-scan” focuses on the perception of present-moment experience associated with the current mind-body status [21]. The specific aim of the “body scan” is to focus attention on successive parts of the body, frequently beginning with a mental scan of the left foot and ending with the top of the head, to become more mindfully aware of the precise bodily feelings and sensations [22]. Alternatively, individuals focus on breathing in and out. In fantasy-guided imagery, the mind is directed to intentionally visualize places, objects, or events that are not externally present with the aim of influencing psychological and physiological states [23]. Thus, given the different focus of attention in the two interventions one could assume that individuals engaging in mindfulness breathing meditation would experience different states of mind than subjects embarking in guided imagery with regard to their perception of time and space correspondingly also exhibiting impact upon their self-perception pattern. 

To empirically address the question of whether different short-term meditative and relaxing settings produce specific or generic-state changes directly after meditation, we conducted a study with individuals moderately familiar with meditation and relaxation techniques. We recruited volunteers in an undergraduate integrative-health-promotion program at the University of Applied Sciences Coburg (CUAS). We assessed and compared subjective time, space, and self as basic constituents of conscious experience during a meditation and fantasy-relaxation intervention session. The two induction methods are different in how attention is directed either to the body (meditation) or to an imagined world (fantasy-relaxation). Since the senses of self, time, and space have been shown to be modulated during meditative practice [13,15,24], we hypothesized that these aspects of conscious awareness should be more strongly affected after a meditation, as compared to fantasy relaxation.

## 2. Materials and Methods

### 2.1. Participants and Procedure

Forty-four second year bachelor students enrolled in the integrative health promotion program at CUAS (37 women, 7 men) aged between 18 and 35 years (mean age: 23.9 years; S.D. = 5.4) participated in this study. The students had already been familiarized with stress reduction, relaxation and meditation techniques as part of their training. Thus, they had some prior exposure to both types of relaxation inductions utilized in this study. They had been introduced to the theory of mind-body techniques and also participated in an 8-week mind-body related stress-reduction program in the foregoing semester, which has been described and evaluated elsewhere [25,26]. The current study was conducted at the end of the fourth semester, after the students had been introduced to different forms of relaxation techniques including mindfulness meditation, walking meditation, fantasy journey, as well as yoga, progressive muscle relaxation, and autogenic training under the supervision of two mind-body experts for the whole semester (NK and TE). They were randomly assigned to two different experimental groups (*n* = 22 in each class). The two groups received the same intervention only in a different temporal order: One group (*n* = 22) started with the mindfulness breathing meditation (t1), and the other (*n* = 22), with the guided fantasy relaxation (t1). The two groups switched interventions (t2) one week later. 

At the first session, before the respective intervention was conducted, participants in each group filled out the three trait questionnaires described below. Then, the guided intervention either mindfulness breathing meditation or fantasy relaxation lasting exactly 10 min was conducted, after which the students filled out the state questionnaire pertaining to their experiences during the intervention. The audio scripts of both interventions are presented in the Appendix A both in the German original version as well as the English language translation. One week later, the same formal set up for the alternate intervention was used and after the intervention, the state questionnaire was again administered. 

All students were informed about the study purpose and asked whether they were willing to take part during a lesson conducted a week before the study without financial compensation. After the second session, students were given the opportunity to further inquire about the aim and hypotheses of the study as well as the theoretical framework. Participation throughout the study was voluntary. Our study was not linked to any kind of examination students had to take to pass the seminar and the experiment took place after the exams had been conducted. The scientific and value-oriented principles as defined and described by CUAS, served as the fundamental ethical frame for our study, where the combined first-hand-experience of research with lecturing in a classroom setting is considered to be a fundamental educational pillar. The project was categorized as an internal review board (IRB) exempt project and according to the ethical guidelines [27] an IRB waiver was obtained and the study number CUAS-SW-NK-002 assigned. The study was conducted according to the ethical principles of the Declaration of Helsinki.

### 2.2. Interventions

Both mindfulness meditation and fantasy relaxation were recorded interventions in the German language that lasted ten minutes and were guided by the same audio-taped female voice (Lea Gerber).

An excerpt from a recorded, commercially available body-scan meditation was used for the meditation intervention [28] for a transcript, as seen in Appendix A. In the meditation recording, participants are asked to first pay attention to their breath and subsequently to several body parts. We transcribed the relevant text passages and had it spoken by co-author Lea Gerber.

The fantasy journey was produced especially for this study (by Lea Gerber and Lucas Adrian) because there were no commercial products available, which fit our needs. We thereby, achieved comparability with the meditation recording in tone, voice modulation, wording, and length. In the recording, an island scene is described with various animals and ships appearing on the scene, but without any real narrative (for a transcript see Appendix A).

### 2.3. Trait Inventories

#### 2.3.1. Barratt Impulsiveness Scale (BIS-11) 

The Barratt Impulsiveness Scale (BIS) consists of 30 items ranging from 1 (very untrue) to 5 (very true) which are grouped into three subscales: non-planning impulsivity, motor impulsivity, and attention/cognition impulsivity [29]. Sample items include “I plan trips well ahead of time.” and “I spend or charge more than I earn.” According to the validated German version, it is recommended to rely more on the sum score than on individual subscales [30].

#### 2.3.2. Zimbardo Time Perspective Inventory (ZTPI) 

The validated German version of the ZTPI has 56 items ranging from 1 (rarely/never) to 4 (almost always) which are grouped into five subscales representing orientation towards the following dimensions: past-negative (“I often think about the bad things that have happened to me in the past.”); past-positive (“Happy memories of good times spring readily to mind.”); present-hedonistic (“I take risks to put excitement in my life.”); present-fatalistic (“Because things always change, one cannot foresee the future.”); and future (“I am able to resist temptations when I know that there is work to be done.”) [31,32]. 

#### 2.3.3. Freiburg Mindfulness Inventory (FMI) 

The FMI contains 14 four-point items with answer categories ranging from 1 (rarely) to 4 (almost always) which evaluate mindfulness on the basis of a two-factor structure [33,34,35]. The two factors are “presence” as ability to attend to the present moment (“I am open to the experience of the present moment”) and “acceptance” as non-judgmental attitude (“I am patient with myself when things go wrong”) [36]. The factor presence is of special interest as it is conceptually discussed as the propensity to attending mindfully to the “here and now”.

### 2.4. State Inventory

#### Inventory on Subjective Time, Self, Space (STSS)

The STSS has previously been used with varying instructions to assess states of consciousness during a real waiting time [37], while watching two different dance performances [38] and during a depth-relaxation meditation [19]. Participants had to fill out five different visual-analogue scales (VAS). (1) The intensity of awareness of the bodily self and (2) space were assessed with two non-verbal pictorial scales containing answer categories ranging from 0 to 6. The questions were: “How intensively did you experience yourself?” and “How intensively did you experience the surrounding space?” Higher scores indicate greater awareness of body and space. (3) 100-mm-line VASs were presented with the following questions: (3) “How intensively did you think about time?” (anchor points: not at all—extremely); (4) “How fast did time pass for you?” (extremely slowly—extremely fast). (5) Finally, a 100-mm line had to be subdivided into three parts (making two marks) representing the degree of orientation towards the past, present, and future during the intervention.

### 2.5. Statistical Analyses

Within-subject differences for the type of intervention (mindfulness meditation, fantasy relaxation), as well as the session number (t1, t2), were assessed using two-sided t tests for the measures of subjective time, self, and space (STSS). Pearson correlations for associations between the three trait questionnaires and the state scale (STSS) were additionally calculated. The false-discovery-rate (FDR) method, a multiple-comparison-correction procedure [39], controls for multiple tests with an initial *p* value set to 0.05. For statistical calculations we used SYSTAT 13 for Windows.

## 3. Results

In comparing the two intervention conditions (meditation, fantasy-relaxation) over the two time points (t1, t2) we performed separate ANOVAs for the seven state variables. In our case we have two groups of participants who either started with meditation and then did the fantasy-relaxation condition (group 1), or performed the interventions the other way around (group 2). According to this logic the main factor ‘group’ should not lead to differences in the dependent variables as it tests for the between-subjects factor group. Subjects in both groups performed in both conditions and thus there should not be an overall effect for the dependent variables. The group × condition interaction reflects the within-subjects’ differences of the two groups starting with a different intervention condition and thus reflecting differences between t1 and t2 (a potential learning effect over time).

The main factor group did not show a significant effect in any of the seven dependent variables (all F < 1; all *p* > 0.1). The groups starting either with meditation or with relaxation scored equally in the questionnaires assessing the two conditions. One significant effect of the main factor condition appeared for the variable ‘Intensity of the sense of self’ (F = 8.1; *p* < 0.007). The interaction group × condition, reflecting differences between t1 and t2 as a learning effect over time, showed four significant calculations, namely for the dependent variable of ‘Intensity of the sense of self’ (F = 10.5; *p* < 0.002), % Sense of past (F = 5.8; *p* < 0.021), % Sense of present (F = 23.7; *p* < 0.001), and % Sense of future (F = 23.1; *p* < 0.001).

For a better understanding of the above effects revealed by the ANOVAs, in Table 1 we present a table of descriptive mean values in the two conditions and corresponding *t* tests for the answers of the conscious-state scale (STSS) filled out after the mindfulness meditation and the fantasy relaxation. Only one difference proved to be marginally significant, reflecting the main effect of condition in the above ANOVA. The sense of self after fantasy relaxation (3.30) was higher than after mindfulness meditation (2.64) (t = −2.6, p = 0.014; not significant after FDR for 7 calculations). 

According to the ANOVA there were several interaction effects of group × condition, reflecting a sequence effect t1 to t2. Each participant took part in both sessions (with one week between sessions). Therefore, in Table 2 we present the analysis of a potential learning effect. According to t tests, four of the seven variables showed a significant change in average responses. First of all, the sense of self diminished between t1 (3.34) and t2 (2.59) (t = 3.0, *p* = 0.004). All three dimensions of time orientation changed. The present orientation increased at t2 (61.3) vs. t1 (42.9) (t = −4.9, *p* < 0.001); accordingly, past orientation decreased at t2 (17.6) as compared to t1 (22.6) (t = 2.4, *p* = 0.019); a similar decrease was seen for the future orientation (t2: 20.8; t1: 34.5) (t = 3.0, *p* < 0.001). 

Which individual trait correlated with the four state changes in meditation/relaxation over time? The question, which personal disposition enhanced the state effects of our interventions can be obtained from Table 3. For example, mindfulness as a trait could have been sensitive for state effects, but correlation coefficients were not related. Two related trait variables were significantly associated: impulsivity and present hedonism. The more impulsive individuals were, the less increase in the present orientation was experienced over the two sessions (r = −0.473, *p* = 0.001), and the more future oriented they were (r = 0.429, *p* = 0.004). Trait impulsivity counters the effects of meditation/relaxation. Similarly, the more present hedonistic individuals were, the less increase in present orientation they experienced (r = −0.420, *p* = 0.004), and a stronger future orientation was observed (r = −0.473, *p* = 0.002).

## 4. Discussions

In this study, we compared the changes observed in the experience of time, space, and self in students in an integrative health-promotion program taking part in both a mindfulness meditation and a relaxation intervention, employing a within-subject design. The main aim of our study was to compare the effects of two different relaxation techniques, mindfulness meditation and fantasy relaxation, in a group of students with basic experience with those induction methods.

The results indicate that there is one significant difference between the two types of interventions, namely the intensity of the self was observed to be stronger after the guided imagery intervention (significant after the ANOVA). Several significant effects were found when comparing the two intervention time points, regardless of intervention type (ANOVA interaction group × condition). For a better understanding of effects, we collapsed the questionnaire answers of 44 subjects at t1 and t2 regardless of the underlying intervention and thus treated the interventions as equal (as indicated by the null results for differences between intervention types). Effects for a sequence effect were found in four of the seven variables: the sense of self, as well as the past and future time orientation, decreased, while presence orientation increased. Individuals had stronger experiences that are typically found in meditative states at t2 than at t1. The decrease of the sense of self, as well as a stronger sense of presence (at the expense of the past and future orientation) is a typically sign of altered states of consciousness in different relaxation techniques [13,15,24]. Individuals get more absorbed in the ‘here and now’ and show less rumination towards past and future events. The state of absorption in the object of meditation and relaxation is also accompanied by a decrease of the sense of self, a sort of self-transcendence.

Effects on long-term stable traits in experienced meditators show a decrease in rumination and mind-wandering, as well as an increase in self-regulation [17,40,41,42]. Transitory meditative-state effects can similarly be described as an increased sense of presence at the expense of momentary rumination in the past and future. Transitory effects in more experienced meditators have also been described as losing of the sense of self and time, effects which are achieved through an increased presence orientation [11,12,15,43]. Our ‘learning’ effect visible as differences between t2 and t1 corresponds with this concept, as an increased present state at the expense of the past and future awareness accompanies a decrease in the awareness of the self. The relaxation intervention and the mindfulness meditation both led to these changes. Past and future ruminations were suppressed while momentarily focusing on the induction of imaginary images. The immersive experience of listening to the imaginative story apparently led to a loss of the sense of self. 

First-order correlations were detected between the variables that were sensitive to experience changes during the intervention and the three trait-questionnaire instruments tapping into mindfulness, impulsivity, and time perspective. There were moderate associations between the two trait variables hedonistic present orientation (ZTPI) and impulsivity (BIS) and the state variables of present and future orientation during meditation. The more impulsive and hedonistic present-oriented the students were as a personality trait, the less presence orientation they experienced during meditation and the more short-term future oriented they were. Short-term future orientation as a state variable should not be confounded with the long-term future orientation as a trait. More impulsive individuals typically are less future-orientated as a general trait, i.e., they tend to make fewer plans for the future as they concentrate on savoring the present, or the very-near future [31]. Future orientation as a state variable in our context refers to the inability to get absorbed in the present-moment relaxation or meditation inductions. Rather, individuals who are more impulsive or present-hedonists are less able to stay in the present moment and instead focus their attention on the near future, the end of the 10 min intervention.

Moreover, to understand these outcomes, one has to differentiate between an impulsive and hedonistic present orientation and present-mindedness as trained in meditation techniques. The former is associated with a strong urge to act at the present moment, whereas the latter is associated with an observational state associated with more self-control [44]. Our results clearly show that individuals who are more impulsively and hedonistically present oriented have a weaker present orientation and a stronger short-term future orientation while meditating. They have less ability to immerse themselves in meditative or imaginative present awareness.

The fact that we did not identify setting-dependent differences between mindfulness meditation and guided imagery after a 10 min intervention in our study suggests that both interventions—at least if applied for this period of time—produce similar effects that have to be considered generic. The fact that meditation and guided imagery showed no differences with reference to the primary research question in our study could be a result of their common relaxation physiology, i.e., relaxation response pathways [45,46]. The relaxation response (RR) is defined by a set of integrated physiological mechanisms and ‘adjustments’ that are elicited when a subject engages in a repetitive or focused mental or physical activity and passively ignores distracting thoughts. Such behaviors seen in meditation, certain forms of prayer, tai-chi/qigong, yoga, autogenic training, and visualization and guided-imagery procedures, etc., are associated with instantly occurring physiological changes that include decreased oxygen consumption or carbon dioxide elimination (i.e., reduced metabolism), and a lowered heart rate, arterial blood pressure, and respiratory rate [47]. These innate processes might also correspond with an altered state of self or self-perception.

Alternatively, time exposure in the specific setting might have been too short to produce specific effects. Given that we found slightly higher scores in intensity of the self in the guided-imagery class, we believe that a longer intervention period might have produced more pronounced effects. We suggest continuing investigations of state-dependent differences attributable to different MBI settings [48,49] to assess the specificity of intervention mechanisms. A general understanding of different induction methods for altering states of mind could enable us to specifically apply meditation and relaxation methods for different clinical purposes and groups [40,41,50]. The core features of altered states of consciousness are antithetical to psychiatric symptoms [51]. They lead to less awareness of the self and time. In many psychiatric syndromes, such as anxiety and depression, individuals show hyper-awareness of the self and of time. If a person is hyper self-aware, the negative affect is high, and time drags. That is why meditation might have positive effects as intervention in such a health-related context [7,25,26,49].

## Figures and Tables

**Table 1 behavsci-09-00087-t001:** Mean (S.D.) values for the conscious-state inventory on subjective time, self, and space (STSS) for the mindfulness meditation and the fantasy-relaxation intervention.

Measure STSS	Mindfulness Meditation Mean (S.D.)	Fantasy Relaxation Mean (S.D.)
Intensity sense of self [0 … 6]	2.64 * (1.1)	3.30 * (1.5)
Intensity sense of space [0 … 6]	4.25 (1.3)	4.09 (1.7)
Intensity sense of time [0 … 100]	33.6 (26.1)	40.2 (24.0)
Speed of time passage [0 … 100]	58.8 (23.6)	58.1 (18.1)
% Sense of past [0 … 100]	20.6 (13.1)	19.6 (13.3)
% Sense of present [0 … 100]	51.3 (22.3)	52.9 (23.7)
% Sense of future [0 … 100]	27.8 (15.5)	27.5 (18.7)

* *p* < 0.05.

**Table 2 behavsci-09-00087-t002:** Mean (S.D.) values for the conscious state inventory on subjective time, self, space (STSS) for the two sessions (t1, t2), regardless of whether mindfulness meditation or the fantasy relaxation was conducted.

Measure STSS	Session t1 Mean (S.D.)	Session t2 Mean (S.D.)
Intensity sense of self [0 … 6]	3.34 (1.2) **	2.59 (1.4) **
Intensity sense of space [0 … 6]	3.95 (1.5)	4.39 (1.5)
Intensity sense of time [0 … 100]	38.0 (2.5)	35.8 (2.6)
Speed of time passage [0 … 100]	58.3 (22.9)	58.6 (19.0)
% Sense of past [0 … 100]	22.6 (14.6) **	17.6 (11.0) **
% Sense of present [0 … 100]	42.9 (24.0) **	61.3 (17.7) **
% Sense of future [0 … 100]	34.5 (19.0) **	20.8 (11.6) **

** *p* < 0.05 significant after FDR adjustment.

**Table 3 behavsci-09-00087-t003:** Pearson correlation coefficients among the variables of the STSS that were sensitive to changes between t1 and t2 and the personality scales of impulsivity (BIS), time perspective (ZTPI), and mindfulness (FMI).

	Intensity Sense of Self [0 … 6]	% Sense of Past [0 … 100]	% Sense of Present [0 … 100]	% Sense of Future [0 … 100]
BIS Sum	−0.057	0.260	−0.473 **	0.429 **
ZTPI present hedonistic	0.026	0.153	−0.420 **	0.447 **
ZTPI present fatalistic	0.115	0.272	−0.241	0.116
ZTPI future	−0.189	−0.180	0.245	−0.179
ZTPI past negative	−0.211	−0.010	0.214	−0.288
ZTPI past positive	−0.073	−0.087	−0.224	0.368 *
FMI presence	0.002	−0.164	0.176	−0.095
FMI acceptance	0.030	−0.165	−0.023	0.175

* *p* < 0.05, ** significant after FDR-adjustment.

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
