# Peer review of "Mindfulness Meditation and Fantasy Relaxation in a Group Setting Leads to a Diminished Sense of Self and an Increased Present Orientation"

_behavsci, 2019, doi:10.3390/bs9080087_

Round 1
Reviewer 1 Report
The current investigation reports an experimental investigation of mindfulness versus a control relaxation technique on subjective experience of time, space, and self.
The authors report no significant, specific treatment effect of the mindfulness treatment on outcome measures, but indicate decreased sense of self, increased present orientation, and decreased past and future orientations.
This study represents a relatively minor contribution to the understanding of mindfulness.
Line 36: "and yet potentially"
change to: and also potentially
Line 76: "The sense of self, time, 76 and space are modulated during meditative practice [6, 27,41]."
Is this a hypothesis?
This paper needs a clear and expanded hypothesis section. The way line 76 reads, it suggests that your research question is already known in the literature and there is no need for your study. What specifically are you testing new that is not known?
Line 85: This seems to be a serious issue with the study. "after the students had been introduced to different forms of relaxation techniques under the supervision of two mind-body experts for the whole semester."
This prior exposure makes the possibility of a confound by prior skill level with mindfulness.
You must assess for statistical differences in the dependent measures at time 1. Only then can you make statements about changes from T1 to T2, because subjects are not naive.
196: The main aim of the study was to compare the effects of two different relaxation techniques.
This aim was not met.
Line 246: "Alternatively, time exposure in the specific setting might have been too short to produce specific 246 effects."
I agree with this assessment. I don't see support in the literature for single session effects between meditation types.
I feel that your discussion does not offer adequate explanation for why the "the sense of self, as well as the past and future time orientation, decreased, while presence orientation increased."
Why does this occur after already 3 quarters of training in "8 week mindfulness based mind-body program?"
Did this program not have a body scan component at all up until this point? Please clarify.
How did you do 8 weeks of "mindfulness...program" without body scan, as it is such a key component of evidence-based mindfulness programs, such as MBSR?
I would need to see a more thorough discussion of the significance of your trait level regressions.
Author Response
Thank you very much for this fair and helpful review, which has been very valuable for improving the manuscript. we have tried to respond to all points. Please find our comments below.
The current investigation reports an experimental investigation of mindfulness versus a control relaxation technique on subjective experience of time, space, and self.
The authors report no significant, specific treatment effect of the mindfulness treatment on outcome measures, but indicate decreased sense of self, increased present orientation, and decreased past and future orientations.
This study represents a relatively minor contribution to the understanding of mindfulness.
In the context of the emerging studies we believe that our study is remarkable as it shows that relative beginners benefit similarly from meditation as well as more general relaxation induction methods. Potentially, differences in induction-types will also appear to be measured with longer lasting interventions (that is, longer practice). Since this type of studies are now emerging at an exponential rate, we think that differentiating effects depending on method, exposure duration is crucial for an understanding of mechanisms. Derived from our study we can say that for beginners the two different induction methods have a similar impact on states of consciousness, i.e. self and time.
Line 36: "and yet potentially" change to: and also potentially
Done
Line 76: "The sense of self, time, and space are modulated during meditative practice [6, 27,41]." Is this a hypothesis?
We refer to the work referenced under 6, 27, 41 (as examples) as time, space, self are the basic constituents of consciousness. Not only according to Kant experience is organized in time and space and converges as an impression related to a self. Meditation as an induction method to alter states of consciousness will therefore have an effect on time, space, self. And this has been shown empirically before.
This paper needs a clear and expanded hypothesis section. The way line 76 reads, it suggests that your research question is already known in the literature and there is no need for your study. What specifically are you testing new that is not known?
Relating to the above response time, space, self are SOMEHOW altered in different forms of relaxation and meditation methods. We therefore test with different subjective scales the amount (quantity) of changes. However, the differential effects across different induction methods are less known. We therefore test two different induction methods: Also less is known how comparable beginners (with just a few hours of prior exposure) experience changes in subjective states after meditation and relaxation. The reviewer is right that we did not state some hypotheses explicitly. This has now been done.
Line 85: This seems to be a serious issue with the study. "after the students had been introduced to different forms of relaxation techniques under the supervision of two mind-body experts for the whole semester."
This prior exposure makes the possibility of a confound by prior skill level with mindfulness.
We would term this ‘confounder’. As stated, the students were exposed “to different forms of relaxation techniques”, that is, a variety of methods, including meditation and fantasy relaxation.
So, students had some prior exposure to both types (see below). We write this more clearly in the revision. The previous statement might have been misleading and has therefore been corrected.
You must assess for statistical differences in the dependent measures at time 1. Only then can you make statements about changes from T1 to T2, because subjects are not naive.
There are indeed good arguments for looking at between-subjects differences for the two conditions. In this case, as the reviewer writes, one subject has only performed one relaxation condition (between-subjects design). Accordingly, we performed a t test for t1 only and compared the two conditions. No significant differences appeared. However, when taking only t1 we have less power and some potential effects might have gone unnoticed that would become evident with more subjects. As we concede, there is definitely room for more research. The types of research questions and studies we and others are conducting are emerging right now. We consider his study to be a part of a process.
196: The main aim of the study was to compare the effects of two different relaxation techniques.
This aim was not met.
And why not? We don’t see an argument here. We opine that we have contrasted the effects of different forms of relaxation techniques on time, self and space perception at two points of measurement in two groups with inverted setting exposure.
Line 246: "Alternatively, time exposure in the specific setting might have been too short to produce specific effects."
I agree with this assessment. I don't see support in the literature for single session effects between meditation types.
Well, good for us. So accordingly we would be the first to do this. That is science, to start a new inquiry. We ourselves have shown how a 6.5 minute meditation-type intervention had an effect on subjective time and emotion (Pfeifer et al. 2016). Linares et al. (2019) also performed a 10-minute meditation and found an intervention effect. The rationale for using a 10-minute time range is that for beginners a longer intervention could become a tedious task and then effects would go away. This rationale is based on our own work. Nevertheless, and only future work may tell us, a longer intervention session, even in beginners, could have a stronger effect. Apart from that, for healthy individuals being exposed to relaxation, 10 minutes’ interventions are frequently recommended as a suitable time frame (Kersemaekers et al, 2018).
I feel that your discussion does not offer adequate explanation for why the "the sense of self, as well as the past and future time orientation, decreased, while presence orientation increased."
We agree that we should have discussed this more thoroughly. We do so now.
“The decrease of the sense of self as well as a stronger sense of presence (at the expense of the past and future orientation) is a typically sign of altered states of consciousness in different relaxation techniques [6, 27, 41]. Individuals get more absorbed in the ‘here and now’ and show less rumination towards past and future events. The state of absorption in the object of meditation and relaxation is also accompanied by a decrease of the sense of self, a sort of self-transcendence.”
Why does this occur after already 3 quarters of training in "8 week mindfulness based mind-body program?"
Our participants were students enrolled in a health promotion bachelor program, where they are familiarized with a set of relaxation and stress reduction techniques. Our participants were 4th semester students who had been introduced to a multi-level stress reduction program including aspects of nutrition, behavioral changes, exercise and relaxation the foregoing semester (Esch& Esch, 2013). In the ongoing semester they had practiced relaxation techniques such as progressive muscle relaxation, yoga, Pilates, autogenic training as well as fantasy relaxation and body scan during the semester.
Did this program not have a body scan component at all up until this point? Please clarify.
The program included one exercise in body scan as well as a fantasy journey, so that students could familiarize themselves with the respective techniques.
How did you do 8 weeks of "mindfulness...program" without body scan, as it is such a key component of evidence-based mindfulness programs, such as MBSR?
See above, both body scan and fantasy journey were part of the structured stress management program in a comparable manner (additional interventions included e.g. introspective diary, walking meditation and breathing exercises), so that there was no unbalanced overexposure of one of the two techniques (i.e. body-scan and fantasy journey).
I would need to see a more thorough discussion of the significance of your trait level regressions.
We agree with the reviewer. We have added a paragraph to the discussion:
“Short-term future orientation as a state variable should not be confounded with the long-term future orientation as a trait. More impulsive individuals typically are less future-orientated as a general trait, they make fewer plans for the future as they concentrate on savoring the present, or very-near future [44]. Future orientation as a state variable in our context refers to the inability to get absorbed in the present-moment relaxation or meditation inductions. Rather individuals who are more impulsive or present-hedonists are less able to stay in the present moment and focus their attention on the near future, the end of the 10-minute intervention.”
Reviewer 2 Report
I've enjoyed reading your paper, comparing the effects of mindfulness meditation and fantasy relaxation on sense of self and time orientation among students of meditative practices. I think it provides some valuable insights about the differences and similiraties between various meditative techniques. However, the presentation of the study can be improved by clarifying the following points:
From the introduction it does not become sufficiently clear to me what the hypotheses and the purpose of the study are. How does this study contribute to current scientific debate and/or to (clinical) practice (this does not become clear until the very last sentence of the paper)? What did the authors expect with regards to the differences between the two interventions?
In relation to the previous comment, it also doesn't become sufficiently clear why the authors chose to study sense of self, time and space as the dependent variables. Also, how do these relate to health, which seems to be an important rationale for the study as evidenced in the first paragraph of the introduction.
Why did the authors choose to conduct t-tests examining the group differences and then correlational analyses on the association with possible confounders (the traits impulsiveness, time perspective and mindfulness), instead of using ANCOVA and/or MANOVA to control for interrelations between the variables and effects of multiple testing?
On p7, lines 230-232, the authors suggest that the correlation between hedonistic present orientation and impulsivity on the one hand and present and future orientation during metdiation of the other hand, reflect a causal relation. What is this assumption based on? After all, correlation is not causation?
In lines 233-245 a connection is made with health again. However, a little more explanation is needed on why an althered stat of self or self-perception correspond with the physiological changes mentioned. Might there not be other characteristics of the techniques discussed that mediate these physiological responses?
Finally, I would recommend having the manuscript checked by a language editor to take out some odd sentence structures and other sentence level mistakes.
Author Response
Thank you very much for this helpful, fair and thorough review, We feel that your review has been very helpful in amending the manuscript! Please see our answers below:
I've enjoyed reading your paper, comparing the effects of mindfulness meditation and fantasy relaxation on sense of self and time orientation among students of meditative practices. I think it provides some valuable insights about the differences and similiraties between various meditative techniques. However, the presentation of the study can be improved by clarifying the following points:
From the introduction it does not become sufficiently clear to me what the hypotheses and the purpose of the study are. How does this study contribute to current scientific debate and/or to (clinical) practice (this does not become clear until the very last sentence of the paper)? What did the authors expect with regards to the differences between the two interventions?
We agree on this point. We therefore have added an explicit hypothesis at the end of the introduction.
“The two induction methods are different in how attention is directed either to the body (meditation) or to an imagined world (fantasy-relaxation). Since the senses of self, time, and space have been shown to be modulated during meditative practice [6, 27, 41], we hypothesized that theses aspects of conscious awareness should be more strongly affected after a meditation as compared to after fantasy relaxation.“
In relation to the previous comment, it also doesn't become sufficiently clear why the authors chose to study sense of self, time and space as the dependent variables. Also, how do these relate to health, which seems to be an important rationale for the study as evidenced in the first paragraph of the introduction.
The senses of self, time and space are the constituents of conscious awareness which become altered in several extraordinary states of consciousness such as in meditation. Therefore we included these measures here. Why the relation to mental health? “The core features of altered states of consciousness are antithetical to psychiatric symptoms. They lead to less awareness of the self and time. In many psychiatric syndromes e.g. related to anxiety and depression, individuals show hyper-awareness of the self and of time. A person is hyper self-aware, negative affect is high, and time drags. That is why meditation has positive effects as intervention in a health-related context dealing with predominant mental health problems such as anxiety and depression. We have added this paragraph to the end of the Discussion.
Why did the authors choose to conduct t-tests examining the group differences and then correlational analyses on the association with possible confounders (the traits impulsiveness, time perspective and mindfulness), instead of using ANCOVA and/or MANOVA to control for interrelations between the variables and effects of multiple testing?
The reviewer raises a point in that we should first report the overall model by integrating the two factors of condition (meditation, relaxation) and time (t1, t2). To not overload the statistics and to concentrate on the state variables we therefore report the ANOVA results first and then add the t tests for clarification.
On p7, lines 230-232, the authors suggest that the correlation between hedonistic present orientation and impulsivity on the one hand and present and future orientation during metdiation of the other hand, reflect a causal relation. What is this assumption based on? After all, correlation is not causation?
In line with the reviewer we present only correlations. This is the cautious approach with a cross-sectional study design. However, we have stable trait variables on the one hand and situational state variables which make a more causal interpretation possible. This is also done before the rich background of studies showing how more impulsive and hedonistic individuals (trait) behave in a certain way in situations (state) as referenced in the work by Zimbardo and Boyd [44]. In our case the trait of being more impulsive and hedonistic leads to less absorption or the feeling of presence during the interventions.
In lines 233-245 a connection is made with health again. However, a little more explanation is needed on why an althered stat of self or self-perception correspond with the physiological changes mentioned. Might there not be other characteristics of the techniques discussed that mediate these physiological responses?
This has been added in the manuscript.
Finally, I would recommend having the manuscript checked by a language editor to take out some odd sentence structures and other sentence level mistakes.
We have had a native English speaker, also a professional, a retired medical doctor, check the whole manuscript.
Round 2
Reviewer 1 Report
I feel as if the requested revisions were completed at this time.